# Fault Diagnostic Methodologies for Utility-Scale Photovoltaic Power Plants: A State of the Art Review

Qamar Navid [1,*], Ahmed Hassan [2], Abbas Ahmad Fardoun [3], Rashad Ramzan [4] and Abdulrahman Alraeesi [5]

1   Emirates Centre for Energy & Environment Research, United Arab Emirates University, Al Ain 15551, United Arab Emirates
2   Department of Architecture Engineering, College of Engineering, United Arab Emirates University, Al Ain 15551, United Arab Emirates; Ahmed.Hassan@uaeu.ac.ae
3   Department of Electrical and Electronic Engineering, Al Mareef University, Beirut 1001, Lebanon; abbasfardoun@gmail.com
4   Department of Electrical Engineering, National University of Computer and Emerging Sciences, Islamabad 44000, Pakistan; rashad.ramzan@nu.edu.pk
5   Department of Chemical and Petroleum Engineering, United Arab Emirates University, Al Ain 15551, United Arab Emirates; a.alraeesi@uaeu.ac.ae
*   Correspondence: qamar.navid@uaeu.ac.ae; Tel.: +971-56-8319571

**Abstract:** The worldwide electricity supply network has recently experienced a huge rate of solar photovoltaic penetration. Grid-connected photovoltaic (PV) systems range from smaller custom built-in arrays to larger utility power plants. When the size and share of PV systems in the energy mix increases, the operational complexity and reliability of grid stability also increase. The growing concern about PV plants compared to traditional power plants is the dispersed existence of PV plants with millions of generators (PV panels) spread over kilometers, which increases the possibility of faults occurring and associated risk. As a result, a robust fault diagnosis and mitigation framework remain a key component of PV plants. Various fault monitoring and diagnostic systems are currently being used, defined by calculation of electrical parameters, extracted electrical parameters, artificial intelligence, and thermography. This article explores existing PV fault diagnostic systems in a detailed way and addresses their possible merits and demerits.

**Keywords:** utility-scale power plants; photovoltaic (PV); monitoring; fault diagnostics

## 1. Introduction

With current rates of market penetration, renewable energy generators are predicted to reduce reliance on conventional fossil fuel power systems by up to 60% by 2020 [1]. Photovoltaic (PV) has witnessed the highest increase in installed capacity among renewable energy generators, with a power generating capacity of 400 GW in 2017, which is 32% higher than 2016 [2,3]. This fast-paced growth is driven by the immense solar irradiance potential worldwide that can generate enough electric power in one hour [4] to meet world energy demand for one year [5].

The drastic decrease in the leveled cost of energy (LCOE) for the standard 10–100 kW PV system, from 1140–14,000 €/kWp from 1990–2017, further increased the consumer acceptance of PV systems [6]. Cell material and technology-dependent PV efficiency of 10.2%, 25.6%, 28.8%, and 44.7%, respectively, were reported for amorphous Si, GaAs, and crystalline silicon and multifunction solar cells in [7,8].

Although PV systems are both scalable and market competitive, they face serious challenges in reliable operation when applied to utility-scale plants due to the frequent occurrence of faults and the time involved in fault clearance [9]. PV generators are prone to faults on the direct current (DC) side of the generator and the alternating current (AC) side of the grid, which is connected via the inverter. Defaults commonly recorded on the DC side include misalignment, interconnection and connection, earth, arc, line to line, bypass

diode, short circuit (SC), open circuit (OC), maximum power point tracking (MPPT), and degradation. In the worst case, the incidence of faults increases as the panels deteriorate over time, reaching up to 0.8% per year [10]. The degradation caused by metal oxidation, connectors corrosions, bus bar discoloration, and rise in PV module series resistance may lead to serious faults and discontinuity of operation [11]. Faults predominantly reported on the AC side include cabling and sudden natural faults [12–14]. The nonlinear output behavior of the PV system renders it extremely difficult to detect the nature and position of these faults. The specifics of each fault reported in the PV system are summarized in Table 1.

To maintain an efficient and reliable power supply, the fault should be diagnosed and cleared with minimum possible delays to avoid plant shut down or damage that may lead to fire occurrence [15].

Several fault monitoring methodologies ranging from manual observation to fully automated diagnosis are currently practiced [16,17]. It is important to determine the exact characteristics and location of faults, along with a well-defined defect clearance process, to preserve the reliability of operations for large PV power plants. Such complexity in fault detection and intervention has invited tremendous interest from the research community to resolve the issue [18].

Utility-scale PV power plants require extreme diligence and precision in fault diagnosis due to their influence on grid stability, which has wider implications compared to stand-alone PV arrays. A general fault diagnostic relies on monitoring the daily performance ratio (PR), comparing the measured PR with the estimated PR applying a statistical approach, and declaring a fault when the variance exceeds the pre-defined threshold value [19,20]. The measured PR varies from the predicted PR based on the nature of the fault, which helps determine the type of fault [21].

The fault diagnostics are broadly categorized into electrical parameters analysis, thermal imaging, and artificial intelligence [22]. Electrical methods depend on the understanding of ideal and non-ideal open-circuit voltage ($V_{OC}$), short circuit current ($I_{SC}$), and maximum power point (MPP) deterministic I-V and P-V curves (MPP). Time-domain reflectometry (TDR) is employed for the time and position of faults in which input and the reflected signal from the PV module are compared. Failure position and type are defined by the signal delay and distortion in waveform shape [23–25].

Microscopic research, which requires experimental work including (i) scanning electron microscopy (SEM) (ii) attenuated total reflectance infrared microscopy (ATIR), and (iii) X-ray micro-tomography, typically locates the DC side faults. The first indication that the PV system has a fault is the drop in DC power output. However, after studying the electrical characteristics of the PV system, it is reported that there is a fault in the system [26].

DC power from the array is slightly below the predicted quantities, as certain electrical faults, for example, failures, often occur inside all arrays. Table 1 shows the most common fault forms in the PV system.

The current article discusses in depth the state of the art of faults in utility PV plants and their identification, using electrical parameters, artificial intelligence, and thermal imaging. Furthermore, this article discusses the most effective and efficient monitoring techniques which are feasible for implementation. This article presents the performance of individual techniques with an emphasis on: (1) methods, (2) software and hardware constraint, (3) capability of fault diagnostic and classification, (4) integration complexity, (5) cost-effectiveness, (6) accuracy and precision. Moreover, a detailed investigation is conducted to investigate the different types of defects in the PV system, their implications for electric and thermal parameters, their disadvantages, and safety challenges.

**Table 1.** Photovoltaic (PV) System faults—classification and localization.

| Fault Type | Sub Faults | Description |
|---|---|---|
| Mismatch Faults | Partial shading [27,28] | Trees existence, overhead supply lines, nearby structures |
| | Uniform irradiance distribution [29] | Non-uniform nature of irradiance in the day |
| | Soiling [30] | Dirt accumulation and droppings due to birds |
| | Snow covering and hot spot [31,32] | Immense change in temperature in tropical regions |
| | Upper ground fault [33,34] | An unnatural ground path with no impedance between the last two PV string modules |
| | Lower ground fault [33,34] | An unnatural path to the ground with null impedance between the second and third modules in PV string through a huge back-feed stream |
| | Series arc fault [35,36] | Current conductors having discontinuity caused by solar disjoint, damage of a cell, connector's corrosion, |
| | Parallel arc fault [35,36] | Breakdown of insulation in current conductors |
| Line to line faults [33,37] | | Short circuit among the two joints with unlike potentials |
| Bypass diode faults [38] | | Short circuit due to wrong connections |
| Degradation faults [39] | | Cell's coating, delamination, yellowing, browning, bubble, and interconnection results in degradation and increases the series resistance |
| Bridging fault [37,38] | | Loose connection between the different joint having different potentials |
| Open Circuit fault [40] | | Connection breaks down between the solar panels |
| Maximum power point tracking (MPPT) fault [41,42] | | Defects in MPPT controller |
| Cabling faults [43] | | - - - - - - - - - - - - - |
| Inverter faults | | Defects in the inverter components such as Insulated Gate Bipolar Transistor IGBT |
| Sudden natural disasters [44] | | Due to storms, natural lighting, etc. |

## 2. Electrical Review

Fault detection and diagnostic approaches based upon electrical characteristics are categorized into analytical redundancy and hardware redundancy. Analytical redundancy is based on a comparison of electrical parameters (output power, voltage, and current) with their corresponding parameters taken from the reference model, while the differential indicates a fault [45–48]. A one diode model (ODM) acts as a reference and the parameters of interest are taken from manufacture datasheets or extracted from performance data applying extraction techniques [49]. The methodology is very sensitive to solar irradiance and thermal parameters which in a sense limits the reliability of the approach [47].

Hardware redundancy involves a comparison of the measured data from several systems functioning under the same settings. By monitoring and analyzing data from each subsystem, abnormalities are determined. The limitation of the method lies in the handling of a tremendous amount of data being gathered and analyzed which necessitates the use of artificial intelligence and specialized data handling techniques. The specific faults detected by the electrical method are discussed below.

## 2.1. Mismatch Faults and MPPT

Mismatch effects in PV arrays are examined in various shading conditions for optimum power-point tracking [50–52]. Although the introduction of MPPT converters improves the output power by 20–30%, this comes at the expense of increased complexity and thereby increases the probability of faults [53]. A PV array setup is established containing 55 modules in each string, their MPP voltage, and insolation level to determine the mismatch faults. A DC-DC converter and charge controller are coupled at an individual string level and operate optimally through a predefined control law to predict the shading effect through the difference in output. The method shows a limitation in terms of the reliability of the converter and a substantial cost added to the system.

The maximum global PowerPoint can be reached by considering voltage bands, which can be easily calculated using simple analytical methods. [54–56]. Calculation of voltage bands associated with maximum peaks is reported in the P-V characteristics of the device by diode number. Compared with Perturbation-and-Observation (P&O) algorithms, the existence of many local maxima in a shady setting makes it difficult to determine the global maximum that is resolved by creating an MPPT algorithm. It should be noted that conventional P&O efficiency is less than that of the MPPT algorithm. Although the proposed algorithms work well, their application is limited to well-known conditions. The technique was improved by developing adaptive MPPT (AMMPT) that relies on power and voltage characteristics. The AMPPT algorithms exhibited reasonable success in tracking the global maximum for various fault scenarios and can be even applied in abruptly changing environments [57].

For a particular model integrated PV and converter (MIPC) system, a model-based, closed-loop technique is implemented to achieve maximum power generation from an incorrect or partially shaded PV system. Cuk DC-DC converters act as bypass converters, and Cuk boost terminals are responsible for power conditioning. The working of MIPC on three PV modules involves the measurement of the radiation intensity incidence on PV modules. MPPT models are applied to determine the optimal operating voltage for each module. This terminal voltage is measured and fed into compensation-based controllers which adjust the converter's duty ratio [58]. The method predicts fault precisely with the whole array being shaded; however, it loses accuracy under partial shading conditions. A prerequisite failure tolerant method for the power management system is introduced [59] under shaded and partially shaded conditions. The difference in estimated and measured output power and voltage ratio is used as an indication of faults.

Distributed MPPT that determines MPP at module level is applied to detect all possible fault types through the sensor network [60]. A prototype in MATLAB (MathWorks, Natick, Massachusetts, USA is built and tested in various defective conditions, such as hot spots, soil, cables, and the deterioration of modules mounted on a rooftop PV System. In a small PV system, the proposed approach fits well but has the least precision in the large-scale PV system. A partial shading fault was determined by employing two-diode and Bishop models in Pspice. According to Bishop, several factors can affect the module output including breakdown voltage, module position, and sequence of bypass diodes. The implementation of the methodology optimizes the performance of the bypass diode and reduced the occurrence of hot spots. The technique can also be applied at array scale, but the accuracy drops as the scale of the PV generator increases due to increased input parameters typical of utility-scale PV power plants [61]. Faults due to fabrication tolerance are a topic of interest [62] due to associated energy loss and their dependence upon the real operational environment. Compared to the performance expected in standard test conditions (STC), the real-time performance of a 1 MW PV plant demonstrates that STC conditions predict less loss than the real conditions outlined in Table 2. It is concluded that the mismatch can significantly affect annual power productivity depending on the operating conditions. These faults are categorized as fabrication errors that worsen with increased solar irradiance [63,64].

**Table 2.** Losses in various working conditions [63].

| Working Conditions | Loss (%) |
| --- | --- |
| A | −0.10 |
| B | −0.08 |
| C | −0.02 |
| D | 0.03 |
| E | 0.13 |
| F | 0.40 |
| STC | 0.35 |
| Weighted | 0.21 |

Interconnection laws are implemented to mitigate partial shading losses arising from fixed, easy-to-predict, and unavoidable shading conditions [65]. Interconnection optimization laws are defined relative to different shading conditions and the methodology of optimal interconnection. Re-ordering of PV arrays in case of shading is identified as a remedy showing a 4–5% power increase [66]. The pre-setting current and voltage, given the number of connected panels in series and parallel PV configurations, implement an automated fault-diagnostics technique. This technique offers reasonable simplification and effectiveness in diagnosing the faults for grid-connected PV systems [67]. Electrical signatures such as current and voltage are employed as fault indicators for partial shading and inverter disconnection. Permanent faults like a short circuit, module, or array disconnection, and open circuits are distinguished from temporary faults like partial shading through stay time in the grid-connected system [68]. The relation between electrical phenomena in the PV system and fire risk analysis is performed following international standards, with a relative safety comparison, to ensure adequate identification of defects in the PV system [69]. A relation is established between the faults and respective safety measurements. Experimental studies were carried out on two PV power plants, under fire due to faults, in the USA. The results indicate that hotspots in protection devices converge in advising the deployment of an insulation monitoring device (IMD) and residual current detector (RCM) devices to prevent fire [70].

Satellite data was used for detection and diagnosis of faults induced by snow and variation in solar radiation at a Swiss laboratory LSA-SAF, and German Aerospace Center (DLR). An alarm can indicate an error that can be used to detect defects due to snow covering on a PV system and other error origins. Solar position yield estimates are also improved [71,72]. A fault diagnosis study was established for PV panels in the UK for domestic purposes. In this study, the sustained zero efficiency, short zero efficiency, shading, and non-zero efficiency faults, and the power loss due to these faults, was calculated. The approach shows a limitation in predicting the cause and type of fault, thereby rendering it difficult to resolve [73]. An analytical technique involving a PV module and obstacle coordinate was developed to predict shading by a nearby object, employing the bypass diode effect and hill-climbing technique. When evaluating PV energy, global and local maximums of shading are considered, and shading effects are established. The model was validated in different cities in Turkey with reasonable accuracy and helped in better estimation of PV system behavior. This can assist in sizing and developing PV generators and as a result PV system performance and reliability issues could be solved [74].

Shading profiles have been further investigated under several arrangements of PV array, e.g., series, parallel, total cross-tied (TCT), and bridge-linked (BL). The electrical parameters inclusive of $I_{SC}$, $V_{OC}$, IMPP, $V_{MPP}$, fill factor (FF), and series resistance are compared for shading and related faults in the MATLAB environment. Deviation of all parameters across all PV array alignments has been used for the prediction of partial shading conditions. Furthermore, by employing the measured parameters, more generalized algorithms are developed to determine several faults by applying artificial intelligence [75]. The influence of partial shading on the efficiency of poly-crystalline and mono-crystalline photovoltaic modules was analyzed [76]. Shadows were investigated around the key

electrical features of the PV modules and the bypass diodes configuration. The results of these experiments show that the power generation of the PV module reduces by more than 30% by shading 50% of a PV single cell. The same results as obtained with a single cell are found in the vertically, horizontally, and diagonally shaded profiles added to the PV module. A data extraction technique is applied through the Lambert W function to extract performance parameters through an iterative process. The validated model through real-time performance is extended to optimize system parameters [77].

An online modeling and monitoring algorithm was implemented in LABVIEW under real-time constraints for grid-connected PV systems. Current, voltage, module surface temperature, humidity, and ambient temperature were recorded. The data was analyzed and compared with the Simulink model to predict different faulty conditions. The algorithm can predict power loss exactly, but it does not decide the nature of the failures [78,79]. Virtual reality was applied to a building-integrated PV system to predict partial shading to obtain best fit electrical parameters without changing the position of PV modules [80,81]. Chimneys in the built environment are considered to be a potential source of partial shading for PV panels. A 14.4% energy loss was recorded under real-time operating conditions due to shading effects. [82]. Soiling faults were predicted through the I-V curve characteristic of soiled and non-soiled PV modules. Each module at maximum power point mode was connected to individual electronic switches to measure IV-characteristics. Glass surface soiling decreases the efficiency of PV modules by up to 20% along with localized cells, resulting in a decreased power output and lifespan even in the presence of a bypass diode [83]. The analysis of the IV-curve in normal and shady conditions helps determine the denormalized error (DE). The first derivative of voltage (DE/DV) provides the detected area of the PV module relative to PV voltage. Several PV modules are observed to be dissipating substantial power when receiving non-uniform irradiance. A summary of mismatch faults and their diagnostic methods is given in Table 3.

**Table 3.** A summary of mismatch faults, respective methods, and tasks.

| Ref | Methods | Purposes | Tasks |
|---|---|---|---|
| [50–56] [60] | Maximum Power Point Tracking (MPPT) Distributed MPPT | Monitoring | Generalized shading conditions. Faults in general |
| [57] | Adaptive MPPT | Monitoring | General faults Partial shading |
| [58] | Model Integrated PV and Converter (MIPC) | Monitoring | Partial shading |
| [71,72] | Satellite Data based technique | Detection and diagnostic | Generalized shading conditions. Shading due to snow |
| [74] | Analytical | Detection | Shading |
| [77] | Lambert W function | Monitoring | Partial shading |
| [83] | I-V curve | Prediction | Soiling |

### 2.2. Short Circuit, Open Circuit, and Asymmetrical Faults

A statistical monitoring system is developed by using ratios of power and voltage. These ratios help to accurately detect internal faults, string faults, and MPPT faults with the time and location [84–88]. The technique shows limitations in predicting short circuit, open circuit, and ground faults. Ground faults, if not detected properly can cause power loss and in extreme conditions can create fire safety issues [89]. Sensor-based and a software interface standard open platform communication (OPC) diagnostic technique is effectively applied but only accounts for voltage and temperature as operating constraints. The algorithm differentiates between open and short circuit faults, ground, and shading effects, but is applicable only for small PV systems [90–94]. A new approach to defect diagnosis was employed for the efficient detecting, locating, and separating of OC from SC defects in the PV system [95]. Metaheuristic optimization techniques were used. The proposed approach

succeeded in diagnosing errors in the physical test system and clearly distinguishes the positions and forms of a defect.

A model has been developed to predict faults through working parameters including incident radiation intensity, heat transfer/loss coefficient, and thermal capacitance. The model's applicability is limited since all the working constraints are not included [96]. A comprehensive statistical diagnosis approach that can measure PV system performance and detect the fault and fault type was introduced [97]. The model parameters of the one-diode model (ODM) are used to foresee different operating constraints such as current, tension, and PV power. The exponentially weighted moving average (EMWA) algorithm uses the comparison of measured and projected values and output as input, to describe the fault and its form on the DC side. Although results obtained by using ODM-EMWA are encouraging, the method has some limitations. Error classification is determined by the current and voltage values of the MPP coordinates. Localized shading does not significantly affect the MPP coordinates, so this form of error cannot be measured using this method. Noise can also affect its performance immensely as its presence in the system degrades the correlated data.

Asymmetrical faults have been detected, and the behavior of the voltage controller studied [98]. The important task was to minimize the voltage oscillation of second-order and third-order current inoculation in the presence of asymmetrical faults in the grid. A comprehensive DC side discrete-time model was considered in the design of the compensator. The results showed that the positive side of the direct axis current was dominated by the average DC voltage, while the negative side of the direct current calculated the secondary order oscillation. The approach is simple and effective and can eradicate asymmetrical faults without interrupting normal operation. Outdoor testing has been carried out [99] and the location of faults is determined by calculating earth capacitance (ESM) and time domain reflectometry (TDM). The first method informs of the disconnection I between modules while the second determines module degradation. A novel automated supervisory technique for fault diagnostics based upon power losses was determined [100] by parameter extraction from real-time data including module thermal characteristics and solar irradiance. Power loss and capture losses on the dc side of the PV system are analyzed. Capture losses are divided into two primary categories: thermal and miscellaneous capture losses ($L_{CT}$, $L_{CM}$). Theoretical boundaries are established, and capture losses are regularly monitored; if losses overshoot the preset values, this will be declared as a fault.

A model-based methodology is defined in the state-space model for integrated DC power sources, such as fuel cells, PV, and lithium-ion batteries. By linearizing at its operating points, a state-space model for hybrid sources is created. The fault diagnostic technique depends upon the parameter of the state-space model. This diagnostic method can detect online faults, power production time increases, and damage to the PV system [101]. A DC simulator is developed in MATLAB that involves a purposive set of faults comparing the actual and pre-determined fault scenarios [102]. AC/DC faults in PV are detected by comparing the absolute performance ratio error (APRE) against predefined values. In a grid-connected PV system, an indicator of the normal/faulty condition is produced. The DC-AC power ratio indicator is used to detect the fault. Locating the position of faults in the string and inverter, DC and power ratios are calculated, respectively. MATLAB environment is used to develop and optimize the fault diagnostics [103]. A tool was developed for monitoring PV system installations and their operations. It comprises a relational database management system RDBMS, which contains a joint data center for all the PV systems. Solar is an application that facilitates PV data (sensors, investors, and meters) transfer to a database management system (DBMS) to give a brief overview of its operations. A couple of grid-connected systems were analyzed by using this application and results show that power loss due to thermal effects is between 15–25% in Australia [104].

A new method for automatic defect detection was implemented with the same array set-up for smooth and defective operations in PV generating systems. To detect the gap in energy generation, this technique depends upon the cumulative distribution range. This

diagnostic method consists of two-steps, namely the "in control state" for eliminating several days with irregular behavior, and the "in control data", in which control charts are constructed for monitoring the energy generation of a different day. Parametric and non-parametric statistical models are implemented to define the limits of the cumulative distribution range using control charts for the data produced from the PV system for detecting outliers. The proposed method works well for a PV system having an identical array within the defined limits; however, the method does not work when array setups differ from each other [105].

A study related to faults and defects in the PV system was conducted in Western Australia where solar radiation and temperature are higher than in other parts of the country. PV systems exhibit abnormalities such as defects, faults, and degradation that were not found in normal operation under normal conditions, resulting in power loss [106]. An online diagnostic and tolerance algorithm was implemented for a remotely installed PV system that actively locates PV cell faults and tolerance in cell faults. It was observed that, by modifying the structure of PV modules, the output power can be increased by 81.31% as shown in Table 4 [107].

**Table 4.** Power loss reduction in the proposed online fault diagnostic system [107].

| Faulty PV Cells | 2 | 6 | 10 | 14 | 18 |
|---|---|---|---|---|---|
| Power loss in proposed system (w) | 0.37 | 1.95 | 1.96 | 3.19 | 4.26 |
| Max power loss in baseline (w) | 1.20 | 3.96 | 5.94 | 7.70 | 8.39 |
| Min Power loss of baseline (w) | 0.89 | 2.68 | 3.41 | 3.76 | 4.79 |
| Max power loss reduction | 81.31% | 50.63% | 67.06% | 58.64% | 49.23% |
| Min power loss reduction | 58.38% | 27.18% | 42.57% | 15.16% | 11.06% |

A real-time diagnostic approach has been developed for a photovoltaic system based on the two-diode model by defining the threshold levels for normal and faulty PVs [108]. The electrical parameters of the faulty conditions are extracted from the respective IV-curves and validated through the field test. Failure is detected by comparing the measured output with the optimum estimated capacity. The residual signal is used for the location of defects.

A new fault monitoring technology has been introduced for the Inverter Matrix Impedance Current Vector (IMICV), and radial distribution multiple PV grid-connected systems are simulated [109]. The results are compared between faulty and normal profiles generated in simulation fault analysis software (ANAFAS). These results show that the method performs well for shunt faults and can be extended to the power industry. Wavelet transformation based on an online fault detection system is used for a grid-connected photovoltaic system (GCPV). Multi-stage wavelet transform decomposition is an effective tool for detecting faults and inverter components. The diagnostic procedure implements the standard deviation normalization of the wavelet coefficient for the detection of faults. The developed algorithms are simpler, and no additional hardware is required [110,111]. A summary of short and open circuit faults and asymmetrical faults is shown in Table 5.

**Table 5.** A summary of short, open-circuit faults, asymmetrical faults, and diagnostic methods.

| Ref | Methods | Purposes | Tasks |
|---|---|---|---|
| [90–94] | Sensor network (OPC) | Diagnostic | Open and short circuit faults Ground faults Partial Shading |
| [95] | Metaheuristic | Diagnostic Localization | Open and short circuit faults |
| [97] | One diode model (ODM)-exponentially weighted moving average (EMWA) | Detection Localization | Faults in general |
| [102] | DC Simulation | Monitoring Detection | A set of faults |
| [104] | Relational database management system (RDBMS) | Monitoring | Thermal faults |
| [105] | cumulative distribution range | Diagnostic Detection | Faults in general |
| [108] | Two diode model | Diagnostic Detection | Faults in general Short and open circuit faults |
| [109] | Inverter Matrix Impedance Current Vector (IMICV) | Monitoring | Shunt faults |
| [110,111] | Wavelet transformation | Diagnostic Detection | Faults in general Inverter faults |

### 2.3. Web and Wireless based Fault Detection Techniques

Fault detection and diagnostic technique was conducted based on a comparison of terminal voltage for regular and defective PV arrays [112]. The location of the fault is determined by the difference of different characteristics (v1, v2, v3). By choosing the optimal operational points, system maintenance costs can be reduced by limiting the number of current and voltage sensors. Alongside diagnostic functionality, an identical controller is used for communication purposes (power line communication), thus making it manageable without any external software. The web-based application enables the end-user to access the whole data through the internet. An online fault monitoring technique was presented [113] for off-grid and on-grid PV systems. Measured and estimated power trends are used to predict the various faults in the system. Multi-sensor fault diagnostic arrangement is implemented at the module level, which enables the operator to discern the faults, their type, and what actions can be promptly taken, depending upon the various parameters such as temperature, current, irradiance, and voltage. Analysis of power ratios indicates the faults in the inverter.

A wireless monitoring and fault detection approach based on a STM32F4DISCOVERY high-performance board along with Bluetooth data for communication was employed to attain [114] various parameters of PV modules such as temperature, current, and voltage, attained through a microcontroller via a DAQ board. The captured data are then analyzed in a MATLAB environment and fed into a simulation model. The fault is declared once a 20% variation is detected between the measured and estimated values. Wireless sensor networking (WSN) is used for fault detection at module level [115] involving open-circuit voltage ($V_{OC}$), open circuit current ($I_{OC}$), short circuit voltage ($V_{SC}$), and short circuit current ($I_{SC}$) of the PV module in a string. Alongside the current and voltage measurements, the sensor network can also detect the bypass event, besides being powered by solar panels. This method is more reliable and can trace the exact fault location accurately. However, the effective cost of the system is much higher, which makes it difficult to implement. The ZigBee protocol is used for communication which effectively solves the cost issue in [116].

Hybrid fault detection and a diagnostic monitoring technique with onboard measuring units is introduced in [117]. Electrical and thermal parameters are measured and compared.

Fault detection and classification ate proclaimed when approaching the pre-defined difference between the thermal and electrical parameters. The technique is improved by an integrated wireless telemetry measuring unit in a solar module [118]. The data obtained from the modules enable the health of the installed PV system to be monitored without interference. The current module status is described by the voltage as a state variable. By controlling the current and module temperature, further modifications can be achieved. Table 6 represents the web-based techniques that are being employed currently for fault diagnostics in the PV system.

**Table 6.** A summary of web-based techniques, respective methods, and tasks.

| Ref | Methods | Purposes | Tasks |
|-----|---------|----------|-------|
| [113] | Sensor network | Monitoring and detection | Early fault detection<br>Faults in general |
| [114] | STM32F4DISCOVERY<br>via data acquisition<br>DAQ board | Detection<br>Simulation | Faults in general<br>Partial Shading |
| [115] | Wireless sensor network | Detection at a module level<br>localization | Short and open circuit<br>Bypass diode |
| [117,118] | Hybrid | Detection and diagnostic | Faults in general<br>String and module level |

## 3. Artificial Intelligence Based Analysis

Artificial Intelligence (AI) is evolving from a research interest into an effective rival to the available counterparts. AI has been widely applied in PV performance modeling and fault prediction. Numerical models are generated from available data, and the data being huge provides insight into various aspects of the system under test. Different training algorithms like Artificial Neural Network (ANN), Fuzzy Logic (FL), Simulated Annealing (SA), Genetic Algorithm (GA), Ant colony (ACO), and Particle Swarm Optimization (PSO) have been developed for PV fault diagnostics under various conditions [119–121]. A Support Vector Machine (SVM) approach along with k-nearest neighbors (k-NN) optimizer was implemented for detecting short-circuiting faults in a PV system. Algorithms comprise four steps. Firstly, the voltage and current comparison between the healthy and faulty PV generator establishes the short circuit condition. The second step is to detect the short circuit in a string by considering its output voltage and the third step is the module state characterization. If the voltage is zero and the current is at the maximum point, a PV panel is used as a short-circuit. This can differentiate between short-circuits for a single module or a whole series. The proposed method for detecting short circuit defects is extremely accurate, but time-consuming [122].

Low irradiance level and efficient maximum power point tracking algorithms make it very difficult to detect line-to-line faults in the two-stage support vector machine, and multi-resolution signal decomposition is implemented for line-to-line fault detection under low irradiance conditions [123]. Considering the current, the voltage of strings, and some label details, a two-phase SVM algorithm was developed. The proposed technique is relatively reliable and economical since only the output current and voltage are needed instead of sensors.

In the MATLAB/Simulink environment for the PV system, an intelligent error diagnostic approach was created. This comprises the Neuro-Fuzzy model PV modules, the six separate parameters I-V characterization, and the Norm test. Daily measurement of the I-V curve by the six attributes determined via the hybrid model is the key concept of this diagnostic process. First of all, the Neuro-Fuzzy PV system is established, and secondly, the normal or defective PV system status is determined by the normalization test. The low computational cost of the developed approach makes it easily extendable to string

and array level. Although the technique shows promise, it becomes complicated when a combination of different PV modules is employed [124–127].

Evidence and the fuzzy logic-based algorithm was developed [128] involving membership functions from experiment data, thus introducing a new design for fault detection that improves the accuracy of the detection method. The data fusion technique removes the uncertainty arising from the interface of arrays and can predict fault locality on a large-scale array. Solar Radiation by Teledetection GISTEL and the fuzzy logic combination is used in the detection of faults in PV involving output DC power and total solar radiation, measuring the satellite image transmission coefficient for every pixel and comparing measured values and profiles. Algorithms determine the faults in the PV system by comparing calculated and measured output power ratios. This approach is simple, accurate, and involves fewer input variables [129].

An intelligent automatic approach for fault diagnostics based upon the Takagi-Sugeno-Kahn fuzzy neural network [130] compares the output power of a normal and faulty PV string under defined boundary limits. An alarm is triggered if the measured value lies out of the defined limit.

A new fault diagnostic approach has been implemented for PV systems using the Xilinx Device Generator (XSG) and an Integrated Software Environment (ISE) in the Field-Programmable Gate Array (FPGA). For predefined working environments, irradiance level, panel temperature current, voltage, and peak I-V characteristic are determined by using the simulation model. The simulated model parameters and data extracted from the field measurements are compared to establish faults. The developed approach is simple, cheap, and can be applied to large scale PV plant [131]. A model-based technique employing ANN for investigating the partial shading in a PV system is discussed [132]. Output current and voltage are estimated by ANN in an inconsistent working environment. Different working variables are considered for the measured and estimated model and comparison reveals the state of the module. In combination with ANN and P&O, the failure classification is carried out by maximum power point monitoring using the boost converters in MATLAB\Simulink. ANN curve fitting is employed for evaluating algorithms that are used to mean square error (MSE). The technique showed success in precise fault detection on a single module as well as on an array scale [133]. Mismatch effects, mainly partial shading in PV array, are studied with multiple ANN layers. The ANN model is trained and designed for the full diagnostic process under different environmental conditions. The first layer of the ANN model detects the partial shading and the other two determine the number of shaded modules from the shading factor [134].

A novel fault diagnostic approach has been developed employing a simulated annealing radial basis function (SA-RBF) with kernel extreme learning implemented in MATLAB environment. The model-based and experimentally based plant parameters are extracted for the RBF based extreme learning machine algorithm training. RBF-ELM parameters can be optimized by using the SA algorithm in a very short time, resulting in training accuracy and precision. The algorithm provides very accurate information about short circuits, health, and mismatch faults [135]. In addition to kernel-based extreme learning regarding degradation faults, short circuit failures, open circuit failures, and partial shade patterns, a framework based on I-V features for factor diagnostics is introduced. From I-V characteristic curves, model parameters and environmental conditions are extracted. A seven-dimensional feature vector is developed, and the model input is considered. An extremely fast kernel extreme learning machine (KELM) is used to automatically construct the fault diagnostic model, which has generalized reliably [136].

A multilayer probabilistic neural network (PNN) approach was executed for fault diagnostic analysis. The approach consists of four steps, namely: model parameters extraction, simulation, validation fault declaration, and network construction involving training and testing. A one diode model parameter is extracted by an artificial bee colony (ABC) algorithm and is validated by co-simulation through PSIM™/MATLAB™. Finally, PNN algorithms are implemented to detect different fault scenarios. The model accuracy is

confirmed through the feed-forward backpropagation ANN technique in noisy as well as noiseless environment [137]. A summary of artificial intelligence techniques is presented in Table 7 and discussed in Section 3.

**Table 7.** A summary of AI techniques, their platform, parameters, and targeted faults.

| Ref | Algorithm | Platform | Parameters | Targeted Faults |
|---|---|---|---|---|
| [122] | Support Vector Machine (SVM) & k-NN | - | current and voltage | Short circuit |
| [123] | Two-Stage SVM | - | current and voltage | Line to line |
| [124–127] | Neuro-Fuzzy Model | MATLAB/Simulink | I-V curve | Mismatch |
| [129] | GISTEL & Fuzzy Logic | - | Solar radiation and DC power | Mismatch |
| [130] | Takagi-Sugeno-Kahn fuzzy neural network | - | Output power | Short and open circuit, partial shading, etc. |
| [131] | Xilinx Device Generator (XSG) & Integrated Software Environment (ISE) | Field Programmable Gate Array (FPGA) | Irradiance level, T, I, V, and peak I-V characteristics | Mismatch, connection, cells, diodes, and modules shunted |
| [132–134] | Artificial Neural Network (ANN) | MATLAB/Simulink | Current and voltage | Mismatch |
| [135,136] | Simulated annealing radial basis function (SA-RBF) & KELM | MATLAB | I-V curve | Short and open circuit, health, and mismatch |
| [137] | Artificial bee colony (ABC) & Probabilistic Neural Network (PNN) | PSIM™/MATLAB™ | Current, voltage, and power | Mismatch |

## 4. Thermal Imaging-Based Analysis

The fault diagnostic techniques discussed above are mainly power loss dependent regardless of the type of the fault. Mostly, utility-scale PV plants comprise sprawls over miles that essentially involve obstacles in measuring, transmitting, and analyzing a tremendous amount of data. An alternate approach involves thermal imaging that resolves the problems by involving online data transmission through drones and analysis through control rooms [138]. Image processing techniques can detect PV faults, namely snail trail, and dusting. Images are captured using visible light cameras which are processed based on feature matching. Operating parameters of interest like height, angle of the camera, weather condition, and sun angle are considered. The technique is capable of detecting a specific type of fault in a controlled manner more quickly and efficiently. The capability of light cameras can be enhanced by using advanced fault diagnostic algorithms and modern sensing tools [139]. The image processing technique is also used to identify thermal degradation of Silicon PV modules caused by induced potential (PID). Thermal patterns of the modules are generated using thermography and compared with the intensity model of electro-luminescence images. Therefore, power loss in PV modules is estimated without disconnecting from the array. This method was applied to a PV plant of 500 kW, having various PID impairment, over one hour [140]. IR spectrum-based image processing is extensively studied and currently implemented in the PV industry due to its reliability [141]. Modules with lower power output and higher temperature are efficiently detected through this technique. It can detect short circuits, fractures in cells, and soldering damage. The surface temperature in the defective PV cell is higher than the nearby normal cell so that no global fault detection comparison is possible. Other parameters like distance and angle also have an impact on IR imaging. To detect faulty cells, the local standard deviation and mean intensity of every panel must be characterized which poses a constraint to this technology, although it can detect up to 97% of PV faults [142].

Image analysis through FPGA based cameras is becoming popular nowadays [143]. Triangulation and terrestrial geo-referencing are two approaches to identify PV faults. Thermal profiles for normal panels and faulty panels are analyzed and results are extracted. Triangulation provides comprehensive knowledge about the changes in thermal activity in the IR spectrum. Its drawback is huge data generation, prolonged processing time, and high computational cost. Terrestrial geo-referencing solves these problems by using a global positioning system (GPS) in the IR region [144]. A simple SLIC super pixel-based clustering technique is employed for the identification of hot spots in a solar panel using thermal image processing. This algorithm saves computational time, automates the fault detection approach, and increases power production and efficiency of the PV system [145]. Non-invasive fault diagnostic algorithms have been developed by coupling IRT with FL. The health of the PV module monitored by IR imaging is classified using a neural network. The technology cannot identify all types of faults and advances in the neural network are anticipated [146].

The RF system is an advance in IR thermography through which images are recorded and transmitted from PV arrays to workstations. The data is processed, and faults are diagnosed [147,148]. A statistical approach is used using IR imaging through drones. Single panel faults are studied first and then an array of the module is subjected to test. This technique consists of the following steps, namely normalization, automated threshold, parameter extraction, improvement, and refinement, for deciding faults in modules [149]. Table 8 provides a succinct overview of the applications of drones and thermal imaging cameras in PV fault diagnostics. IR imaging is an excellent tool for fault detection in the photovoltaic system. However, improvements are needed to make the technology more efficient, accurate, agile, and inexpensive for market penetration.

**Table 8.** Unmanned aerial vehicle and cameras.

| Drone | | | Camera | | | | | | |
|---|---|---|---|---|---|---|---|---|---|
| Reference | Device Model | Propulsion | Device Specification | Resolution | Range Specification | Object Temperature Range | Precision | Thermal Sensitivity | Weight |
| [150] | dji S1000 | Electric | GoPRo Hero 3 Optris PI 450 + recorder | 1920 × 1080 382 × 288 | 7.5–13 μm | −20 °C to 100 °C 0 °C to 250 °C 150 °C to 900 °C | ±2 °C or ±2% | 0.040 K | 76 g 380 g |
| [151] | Nimbus EosXi Nimbus PLP-610 | Gasoline Electric | Thermoteknix MicroCAM 640 Nikon1 V1 HD | 640 × 480 3906 × 2606 | 8–12 μm | | | 0.060 K | 74 g 383 g |
| [152] | Nimbus PLP-610 | Electric | Nikon1 V1 HD | 3906 × 2606 | | | | | 383 g |
| [144] | Condor AY 704 | Electric | Optris PI 450 | 382 × 288 | 7.5–13 μm | −20 °C to 100 °C 0 °C to 250 °C 150 °C to 900 °C | ±2 °C or ±2%. | 0.040 K | 320 g |
| [147,148] | | Electric | Flir A35 | 320 × 256 | | −40 °C to 160 °C −40 °C to 550 °C | ±5 °C or ±5% | 0.05 K | 200 g |
| [149] | DaVinci Copters ScaraBot X8 | Electric | GoPRo Hero 3+ Optris PI 45 | 1920 × 1080 382 × 288 | 7.5–13 μm | −20 °C to 100 °C 0 °C to 250 °C 150 °C to 900 °C | ±2 °C or ±2%. | 0.04 K | 76 g 320 g |

## 5. Conclusions and Suggestions

Dwindling supply of fossil fuels and environmental impacts associated with power generation through these sources has prompted the necessity of renewable energy sources, of which photovoltaic is the most promising technology. Contemporary PV systems go from small scale plants for domestic usage to commercial-scale grid integrated plants. The in-situ problems of PV, such as short-circuit and soiling, need to be observed to ensure the smooth and uninterrupted operation of PV plants. Currently available fault-detection technology focuses on electrical, artificial, and thermographic parameters. All technologies have some merits and demerits. For instance, a reduction in the power output of the PV plants can indicate a problem in the plant-based on electrical parameters. However, it is hard to identify the exact location to rectify the problem. Artificial intelligence is another tool for fault prediction which is based on weather data, on-site data, and fault forecasting. The technology needs further advancement as it is time-consuming and incurs much computational cost. The processing and training of the fault diagnostic and detection model is another real-time constraint. Thermography is the most efficient way of PV fault detection which can point out localized faults with the help of unmanned vehicles. The limitation of the thermography technique is its inability to classify faults that can be too dangerous for plant safety and operation. Moreover, thermographic performance may be affected by various atmospheric constraints, such as solar irradiance, dust, angle of incidence, partial shading, and higher atmospheric temperatures.

Based on this analysis, an automatic fault diagnostic and detection system is required, which can be easily integrated with the PV system at the module or string level. The diagnostic system should consist of a sensor network that accounts for important PV parameters and will be used in conjunction with AI techniques to detect faults and their location. The ZigBee protocol can be implemented to ensure the safety of the entire system and plant by keeping in view communication with aspects of the diagnostics system.

**Author Contributions:** Q.N. gathered data and prepared the original draft, A.H. supervises and advises the overall research work and review the original draft. A.A.F. conceptualizes the research work, R.R. edited the draft, A.A. review and finalize the draft. All authors have read and agreed to the published version of the manuscript.

**Funding:** Emirates Centre for Energy and Environment Research–31R106.

**Institutional Review Board Statement:** Not applicable.

**Informed Consent Statement:** Not applicable.

**Data Availability Statement:** Not applicable.

**Acknowledgments:** The research was supported by the "United Arab Emirates University" under "Emirates Centre for Energy and Environment Research", grant number 31R106".

**Conflicts of Interest:** The authors declare no conflict of interest.

## Abbreviations

**Symbols**

| | |
|---|---|
| I | Current |
| $I_{SC}$ | Short Circuit Current |
| $I_{MPP}$ | Maximum Power Point Current |
| $L_{CM}$ | Miscellaneous Capture Losses |
| $L_{CT}$ | Thermal Capture Losses |
| FF | Fill Factor |
| P | Power |
| V | Voltage |
| $V_{OC}$ | Open Circuit Voltage |
| $V_{MPP}$ | Maximum Power Point Voltage |

**Acronyms**

| | |
|---|---|
| ABC | Artificial Bee Colony |
| AC | Alternating Current |
| ACO | Ant Colony |
| AI | Artificial Intelligence |
| ANN | Artificial Neural Network |
| AMPPT | Adaptive Maximum Power Point Tracking |
| APRE | Absolute Performance Ratio Error |
| ATIR | Attenuated Total Reflectance Infrared Microscopy |
| BL | Bridge-Linked |
| DBMS | Database Management System |
| DC | Direct Current |
| ELM | Extreme Learning Machine |
| ESM | Earth Capacitance Measurement |
| EMWA | Exponentially Weighted Moving Average |
| FL | Fuzzy Logic |
| FDOG | First Order Derivative of Gaussian |
| FPGA | Field Programmable Gate Array |
| GA | Genetic Algorithm |
| GCPV | Grid-Connected Photovoltaic |
| GPS | Global Positioning System |
| IRT | Infrared Thermography |
| PID | Induced Potential |
| IMD | Insulation Monitoring Device |
| ISE | Integrated Software Environment |
| IMICV | Inverter Matrix Impedance Current Vector |
| KELM | Kernel Extreme Learning Machine |
| LCOE | Levelized Cost of Energy |
| MSE | Mean Square Error |
| MIPC | Model Integrated PV and Converter |
| MPPT | Maximum Power Point Tracking |
| NE | Normalized Error |
| OC | Open Circuit |
| ODM | One Diode Model |
| PR | Performance Ratio |
| P&O | Perturbation-and-Observation |
| PV | Photovoltaic |
| PNN | Probabilistic Neural Network |
| PSO | Particle Swarm Optimization |
| RF | Radio Frequency |
| RCM | residual current detector |
| RDBMS | Relational Database Management System |
| SA | Simulated Annealing |
| SC | Short Circuit |
| SEM | Scanning Electron Microscopy |
| STC | Standard Test Conditions |
| SVM | Support Vector Machine |
| SLIC | Simple Linear Iterative Clustering |
| SA-RBF | Simulated Annealing Radial Basis Function |
| GISTEL | Solar Radiation by Tele detection |
| TCT | Total Cross Tied |
| TDR | Time-domain Reflectometry |
| XSG | Xilinx System Generator |
| WSN | Wireless Sensor Networking |

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
