# Peer review of "Fault Diagnostic Methodologies for Utility-Scale Photovoltaic Power Plants: A State of the Art Review"

_sustainability, doi:10.3390/su13041629_

Round 1
Reviewer 1 Report
Dear all authors,
firstly, thank you for your manuscript and your interest in research issues of fault diagnostic methodologies for utility-scale PV power plants. I have gone through your manuscript. It presents useful review of mentioned issue above and uses extensive list of the references. Basically, the manuscript is well organized and is in accordance with Sustainability journal template. I agree that it brings the confrontation of main information/statements/facts etc. However, in my opinion, manuscript is not complete critical summary of current research efforts as can be expected from a review. Therefore, the real scientific contribution of the manuscript is not conclusively recognized. Yes, it brings the comparations of methods and their approach i.e. through tables, on the other hand, some parts, let say, some assessed analysis, are only commented generally without provided results confrontations. Please, could you make more precise comparison of the achieved numerical results across individual publications in detail, because a lot of information is general. I am convinced that in case of publishing in the scientific journal, it should provide extensive critical discussion, mainly numerical results or method limits in more detail in context of the state-of-the-art.
Please, I have also some remarks that should be taken into consideration, namely:
- check the meaning of presented values in the introduction - i.e. Number_Unit = noun (x_%) vs. NumberUnit = adjective (x%)
- quantities/symbols should be in italics
Yours sincerely
Author Response
Dear Reviewer,
Please find the response file in the attachment. Many thanks

Reviewer 2 Report
- The font type of symbols can be unified in the form of Italic form to discriminate between symbols and units.
- The references [2,3] can be accurately referred to the following reference: Trends 2019 in Photovoltaic Applications: Survey Report of Selected IEA Countries between 1992 and 2018, Report IEA-PVPS T1-36:2019, 2019. https://iea-pvps.org/wp-content/uploads/2020/02/5319-iea-pvps-report-2019-08-lr.pdf.
- The authors can further figure out and highlight the important findings from the reviewing process.
- Some errors are highlighted in fluorescent as attached manuscript. The authors could double-check the correctness before re-submission.

Author Response

(The authors gave the same response as above.)

Round 2
Reviewer 1 Report
The manuscript has been improved according to the reviewer comments